# Molecular Characterization of Arabinoxylan from Wheat Beer, Beer Foam and Defoamed Beer

**DOI:** 10.3390/molecules24071230

**Published:** 2019-03-29

**Authors:** Jie Li, Jinhua Du

**Affiliations:** College of Food Science and Engineering, Shandong Agricultural University, Tai’an 271018, Shandong, China; lijie_amy@163.com

**Keywords:** arabinoxylan, ethanol fractionation, monosaccharide, avDP, molecular weight, wheat beer

## Abstract

This research was to explore the distribution and some molecular characterization of arabinoxylan in wheat beer (B), beer foam (BF) and defoamed beer (DB) because of the crucial influences of arabinoxylan on wheat beer and its foam. The purified arabinoxylan from B, BF, and DB were fractionated by ethanol of 50%, 67%, 75%, and 80%. The monosaccharide composition, substitution degree (Ara/Xyl ratio, A/X), and average degrees of polymerization (avDP) of arabinoxylan were investigated. Molecular weight and microstructure were also involved in this study by GPC-LLS and SEM, respectively. Under the same ethanol concentration, the arabinoxylan content in the BF was higher than the other two, respectively, and it was precipitated in BF fraction with 50% ethanol which accounted for 80.84% of the total polysaccharides. Meanwhile, the greatest substitution degree (A/X) and highest value of avDP of the arabinoxylan was found in all beer foam fractions regardless of the concentration of ethanol used. The average degrees of polymerization (avDP) of arabinoxylan displayed a significant difference (*p* < 0.05) among B, BF, and DB. Furthermore, arabinoxylan presented varied microstructure with irregular lamellas and spherical structures and the weight-average molecular weight (Mw) of arabinoxylan showed the lowest values in BF, while the largest values were shown in DB. Therefore, arabinoxylan was more accumulated in beer foam, especially in 50% ethanol, characterised by greater value of A/X and avDP, as well as lower Mw. It was suggested that the arabinoxylan played important roles in maintaining wheat beer foam characteristics.

## 1. Introduction

As one of the oldest alcoholic beverages, beer is the third most consumed beverage following water and tea in the world [1]. As is known, wheat beer is a specialty beer accompanied with top-fermentation, outstanding flavour, unique foam and mouthfeel [2], where the percentage of wheat typically has to be at least 40% of the grist bill. As a predominant hemicelluloses based on the whole wheat kernel [3], about 1.5–2.5% of arabinoxylans could be found in the endosperm, while arabinoxylan is one of the main non-starchy polysaccharides of the cell walls, which account for about 66% of total polysaccharide in endosperm cell walls [4,5]. However, as a principal constituent in the wheat cell wall [6], arabinoxylans may cause many problems because the increased amount of wheat used for brewing in wheat beer could negatively influence the wheat beer qualities, such as the haze formation [7], high wort viscosity [8], and the general sensory profile. Previous studies on arabinoxylans in beer were only confined to their contents [9,10], variations during malting, mashing [11,12], and brewing [7,9]. Little is known about the molecular characteristics and distribution of arabinoxylans among different parts of wheat beer.

As one of the major components of cereal endosperm cell walls, arabinoxylan mainly consists consisted of arabinose and xylose [13] and its chemical structure is based on a chain of linear β-(1-4)-D-xylopyranose units, which can be substituted with α-L-arabinofuranose in the O-2 or the O-3 position or both [14]. Arabinoxylan from different parts of wheat kernel differed in their structures. The ratio of arabinose to xylose (A/X) is an important parameter for arabinoxylans and is found to be 0.5–0.6 in wheat endosperm [15]. Furthermore, it was shown that this compound possessed several benefits for human health. According to previous reports, arabinoxylan could enhance immunomodulatory activity [16] and antioxidant activity [17], as well as postprandial metabolic capacity, which helped to decrease the postprandial serum glucose, serum insulin and plasma total ghrelin [18,19]. It was also used as a prebiotic acting on inflammatory bowel disease, rheumatoid arthritis, and type I diabetes [20].

The main objective of this study was to investigate the properties of arabinoxylans including the monosaccharide composition, molecular characteristic, the ratio of arabinose to xylose and its polymerization degree and microstructure in wheat beer, beer foam, and defoamed beer in order to clarify the influences of arabinoxylan on wheat beer and beer foam deeply and provide guidance for wheat beer brewing. Gradient ethanol precipitation was used to purify the arabinoxylan. The composition of monosaccharides and polymerization degree were examined here by gas chromatograph (GC). Meanwhile, the microstructure of arabinoxylan was observed by scanning electron microscopy (SEM), and molecular characterization was investigated by Gel permeation chromatography–multi-angle static light scattering (GPC-MALLS).

## 2. Results and Discussion

### 2.1. Physico-Chemical Analysis

The total solid yields of B/BF/DB are 31.88 ± 0.37%, 32.93 ± 0.72%, and 31.83 ± 1.10%, respectively, from beer, foam and deformed beer. The moistures in these fractions (B/BF/DB) are approximately 4.0% and protein contents are around 14–15%. The contents of ash are 1.0–1.3%. The results demonstrated no significant differences among the yields and chemical compositions of the three samples.

The composition of monosaccharides in the powders of wheat beer, beer foam and defoamed beer are shown in Table 1. The dominant monosaccharide was found to be glucose in all the three materials, which was more than 500 mg/g, followed by xylose and arabinose. The highest content of glucose actually referred to the sum of monomer glucose and glucose hydrolysed from glucan and dextrin because they were determined by GC-FID. Lower levels of galactose and mannose were also detected. It is noted that the contents of glucose, arabinose and xylose are slightly higher in beer form than those in defoamed beer. The same is true with the total sugar content.

### 2.2. SEM Assay

The purified polysaccharide fractions after gradient ethanol precipitation and lyophilization yielded odorless powders with distinctly different microstructure according to the ethanol concentration they were precipitated with (Figure 1). It is remarkable that B50/BF50/DB50 samples are flaky and slightly off-white in colour. When examined under SEM, the samples contained mostly irregular lamellar or fibrous structures (Figure 1A). However, the fractions precipitated at higher ethanol concentrations presented as denser powders in light to dark brown colours (Figure 1B–D). The SEM images of these samples showed distinct spherical particles interconnected by fibrous materials. It was noticed that the samples of B75/BF75/DB75 contained the least fibrous structure and larger spherical particles compared to the samples of B67/BF67/DB67 and B80/BF80/DB80.

Zhou et al. performed some researches on the intracellular polysaccharides from *Phellinus igniarius*, which were precipitated by different concentrations of ethanol [21]. Similar structures of uniform spherical shape and large flakes were observed for polysaccharides precipitated from different ethanol concentrations. Therefore, our results confirmed that ethanol with different concentrations could precipitate non-starch polysaccharides with various appearances and microstructures depending on the composition, structure and molecular weight of polysacchrides

### 2.3. Monosaccharides Composition Analysis

The yields and protein analysis of the four arabinoxylan fractions are summarized in Table 2. It can be seen that the highest yields of around 20% were obtained at 67% and 75% of ethanol precipitates, respectively, but there is no significant difference among the different parts of beer in the same fraction. The contents of protein are less than 10% in the ethanol precipitates under the treatment of enzymatic hydrolysis. B67/BF67/DB67 get the highest protein contents of 10.31 ± 1.30%, 8.38 ± 0.17%, and 7.94 ± 0.05%, respectively. Kang et al. reported the similar results of Gatiflia SD fractions, which were also obtained by gradient ethanol precipitation, suggesting that the amount of protein might be influenced by the different solubility of the fractions [22].

The monosaccharides of arabinoxylan purified from beer, beer foam, and defoamed beer were analysed by GC-FID. As exhibited in Table 2, in 50% ethanol precipitates, the contents of arabinose and xylose account for about 80% of total monosaccharides. This result indicates that the fibrous structures of 50% ethanol fractions shown in Figure 1A can be attributed mainly to the arabinoxylan molecules. While the contents of galactose were relatively low and even glucose was not detected in this fraction. Surprisingly, mannose presented a relatively high level of over 87 mg/g in the precipitates of 50% ethanol which accounted for more than 17% of total monosaccharides. Liepman et al. considered that mannose in polysaccharide was present as a group of a β-1, 4-linked backbone or a combination with glucose residues [23]. Compared with B50/BF50/DB50, the contents of arabinose in B67/BF67/DB67 showed a little decrease, while xylose dropped by half. Mannose significantly decreased to around 9.22–10.65 mg/g in 67% ethanol precipitate, while galactose demonstrates the highest content of 42.61–55.86 mg/g among the four fractions of ethanol precipitate. In addition, glucose appeared with 15.37–23.72 mg/g in 67% ethanol precipitate. However, in the 75% and 80% ethanol precipitates, glucose was the dominate monosaccharide, whereas the contents of arabinose, xylose and galactose decreased markedly. 

The high glucose contents in the 75% and 80% ethanol fractions could be due to the remaining starch and β-glucan that escaped from the enzyme hydrolysis. Verwimp et al. also reported that the same results which showed the isolated arabinoxylan still contained a large level of starch despite the amylolysis in the isolation procedure [24]. This results seems in good agreement with the SEM image in Figure 1C which appeared like undigested starch granules. Thus it can be concluded that arabinoxylans in wheat beer were mostly precipitated by 50%–67% ethanol. Whereas, other polysaccharides in wheat beer required higher ethanol concentration to precipitate. Kang et al. reported the study on polysaccharide of gum ghatti precipitated by gradient ethanol and showed the similar results on monosaccharides [22]. It is assumed that the concentration of ethanol can influence the polarity of the fractions, leading to the different solubility of polysaccharides

The distribution of arabinoxylan in wheat beer and beer foam is presented in Table 3, and the arabinose-xylose polymers (AXP), arabinose-galactose polymers (AG), mannose polymers (MP), galactose polymers (AG) and glucose polymers (GP) are analysed. AXP and MP show an increasing trend while GP decreases under the treatment of enzyme and ethanol precipitation. The contents of AX, MP, AG, and GP were calculated by monosaccharide contents which were determined by GC-FID. Therefore, it is concluded that enzymatic hydrolysis and ethanol precipitation could effectively help to extract polysaccharides.

Arabinoxylan exhibits the highest content in 50% ethanol precipitate, demonstrating more than 300 mg/g and the biggest one of 372.49 ± 1.54 mg/g is in BF50. Additionally, the substitution degree of arabinoxylan (A/X) shows the lowest level with 0.50–0.54 and average degrees of polymerization are 961.58 ± 5.67, 1065.22 ± 3.83, 625.11 ± 1.62, respectively. Hwoever, the largest values of A/X and avDP appeared in the beer foam fraction under 50% ethanol precipitation. A similar trend was also found for arabinoxylan fractions of rye flour with the highest of 450 in 50–60% ethanol precipitate [24]. However, AG is in a low level of around 10 mg/g in 50% ethanol precipitate. Furthermore, the content of MP is the second highest in 50% ethanol precipitate with 78.64–87.60 mg/g and it is reported the first time in the literature. But GP was not detected by GC-FID in B50/BF50/DB50 while a little β-glucan and dextrin was detected by a Megazyme enzyme reagent kit due to the high sensitivity of the method. Due to the absence of glucose in 50% ethanol precipitate, it could be speculated that mannose mainly exists in the form of mannan rather than glucomannan in this fraction. In 67% ethanol precipitate, AX showed a decreasing trend to around 200 mg/g while AG increased to more than 64.60 mg/g. A/X value was 0.68–0.70 and avDP decreased sharply to 49.79–60.29. However, the largest value of A/X (0.70 ± 0.01) and avDP (60.29 ± 1.01) were also appeared in beer foam fraction in 67% ethanol precipitates. MP decreased but DG appeared. However, the ratio of mannose to glucose is close to 1:2 (Table 2), it is speculated that its existence form may be glucomannan in 67% ethanol precipitate [25]. In 75% ethanol precipitation, the content of DG was the highest while AX shows the second level of 20–30%. With the increasing ethanol concentration, avDP showed a sharp decrease to around 58.60–89.96 in the 75% ethanol precipitation, but it was still higher than the raw material (B/BF/DB). Meanwhile, avDP also showed the same trend with A/X which was higher in the beer foam (BF50, BF67, BF75) than any other two fractions (B50, B67, B75, and DB50, DB67, DB75).

Therefore, we could know that arabinoxylan is mainly collected in the 50% ethanol precipitates with the highest value of A/X and avDP, especially in beer foam (Table 4). Surprisingly, MP was first found and shows a relatively high percentage in B50/BF50/DB50, but AG indicated a higher percentage in 67% ethanol precipitates. What’s more, with the increasing ethanol concentration, arabinoxylan showed a decreasing trend while DG was opposite. MP in 50% ethanol precipitate and AG in 67% ethanol precipitate will be worth to investigating their origin and existence manner in the next research. We found that A/X value of beer foam was the highest in any fractions. In our previous studies, the A/X value showed around 0.50 in the malting, wort preparation and brewing [9,11]. However, with increasing ethanol concentration, arabinoxylan was precipitated accompanied by increasing A/X ratio (Table 3), and this was also observed in AX fractions of wheat flour [26] and barley flour [27]. Zhang et al. and Izydorczyk et al. reported that A/X had different value from various materials because of the manner of arabinose residue substitution in xylan backbone of the relative proportions, sequence of the various linkages between these two sugars (xylose and arabinose), and presence of other substituents [28,29].

### 2.4. Analysis of Molecular Weight

The analysis of weight-average molecular weight (Mw) distribution was detected by GPC-MALLS with the detector signals of LS and differential refractive index (Figure 2). The recovery rate of all samples in GPC-MALLS was assumed to be fully recovered. It was shown that Mw were significantly different among the samples. The molecular characteristics of fractions precipitated by gradient ethanol are analysed and presented in detail in Table 5. In B50/BF50/DB50, it was shown as 94.56%, 93.56% and 92.64% of total mass after analysing with Mw of 482.40 kDa, 472.20 kDa, and 496.00 kDa, respectively. In the precipitates of 67% ethanol, two peaks signal could be separated for 243.00 kDa, 233.00 kDa, and 203.20 kDa with the proportion of 23.40%, 21.54%, and 25.55%, and another peaks of 21.77 kDa, 20.79 kDa, and 22.26 kDa, which were associated with the proportion of 76.60%, 78.46%, and 74.46%. In addition, B75/BF75/DB75 possessed more than 98% mass fraction of the total fraction and Mws were 4.514 KDa, 3.275 KDa, and 5.163 KDa, respectively. Mw/Mn (number-average molecular weight), a polydispersity index which was closer to 1.000, indicated that the factions were more homogeneous. Mw/Mn in peak 2 of 50%/67%/80% ethanol precipitates was around 1.000, and it declared that the length of chain in these fractions were uniform and concentrated. However, Mw/Mn in 50% ethanol precipitate were in the range of 3.123–5.628, which explained the wide and inhomogenous peak of these fractions. Thus, it was found that the macromolecular arabinoxylan (more than 200 kDa) was mainly collected in 50% and 67% ethanol precipitates, indicating a decreased Mw trend with the increasing ethanol concentration.

## 3. Materials and Methods

### 3.1. Materials

Wheat beer was made as previously described in our research [9]. Barley malt (0.51% WEAX) and wheat malt (1.08% WEAX) with the ratio of 1:1 were milled and mixed with water, the mixed mash then underwent a protein rest, saccharification, lautering, wort boiling, hopping, and a cooling phase in sequence. Hop pellets were added at the beginning of boil. The boiled wort was separated from the hop waste and cooled to 20 °C and oxygenated. A 10 °P wort was transferred into a fermentation vessel (Shandong Taishan Beer Co. Ltd, Tai’an, China), pitched with *Saccharomyces cerevisiae* WL 300 and fermented at 20–22 °C. Once the diacetyl level reached the limiting value, the green beer was immediately cooled to 0 °C and then entered maturation to wheat beer for analyse.

Ethanol was purchased from Tianjin Kaitong Chemical Reagent Co. Ltd. (Tianjin, China). α-Amylase (EC 3.2.1.1; E-BLAAM) and protease (Subtilisin A, EC 3.4.21.62; E-BSPRT), lichenase (endo-1,3:1,4-β-D-Glucanase, EC 3.2.1.73; E-LICHN), and amyloglucosidase (EC 3.2.1.3; E-AMGDF) were all obtained from Megazyme International Ireland Ltd. (Wicklow, Ireland). Monosaccharides standards, including L-arabinose, D-xylose, D-mannose, D-galactose, D-glucose (purity at least 99%), were ordered from Sigma-Aldrich (Shanghai, China). All other chemicals (reagents and solvents) were of analytical grade unless otherwise specified.

### 3.2. Preparation of Samples

#### 3.2.1. Separation of Beer Foam

The separation of beer foam was performed according to our previous study [30]. A bottle of beer was poured into a separating funnel at a constant speed along the edge of funnel (diameter = 20 cm) to generate foam. The upper foam and lower liquid were collected as beer foam and defoamed beer, respectively. The sampled beer, beer foam and defoamed beer were centrifuged at 4500 rpm for 15 min before further analysis. The obtained beer foam, defoamed beer, and original beer were concentrated to remove ethanol using rotary evaporators (RE-52AA, Shanghai Biochemical Instrument Factory, China) and then lyophilized in a freezing-dryer (TF-FD-18S, Shanghai Tian-Feng Industrial Co. Ltd., China) until dry powders were formed. Then the powder was collected and labeled as B (beer), BF (beer foam) and DB (defoamed beer) and thereafter stored in –20 °C. The yield of the B, BF and DB fraction in the manuscript referred to the ratio of B/BF/DB powder to B/BF/DB liquid after freeze-drying. The calculation formula is as follows:(1)$$\mathrm{Yield}{\;}^{1}\;(\%)=\;\frac{\mathrm{weight}\;\mathrm{of}\;B/\;\mathrm{BF}/\;\mathrm{DB}\;\mathrm{powder}}{\mathrm{weight}\;\mathrm{of}\;B/\;\mathrm{BF}/\;\mathrm{DB}\;\mathrm{liquid}}\times 100\%$$

#### 3.2.2. Purification of Arabinoxylan

Enzymolysis was conducted according to the Megazyme manuals to purify arabinoxylans. The manuals’ weblinks were as followings: α-Amylase [31], protease [32], lichenase [33], amyloglucosidase [34]. Firstly, fat was removed from the powder of B/BF/DB by vibrating extraction for three times with n-hexane (1:5, *w*/*v*) under the condition of 50 °C, 2 h, 200 rpm, and then the defatted sample of B/BF/DB was dried for enzymolysis [35]. The enzymatic hydrolysis procedures were performed as follows.

##### Step One

A 20% aqueous solution of B/BF/DB was made with the original pH of 4.53/4.56/4.56, and then sodium hydroxide solution (NaOH, 6 mol/L) was used to adjust pH to 6.00 for enzymolysis. A 100 mL aqueous solution of B/BF/DB in a 500 mL bottle was incubated in a 60 °C water bath, then 600 μL of α-amylase and 300 μL of lichenase were added into the bottle by shaking for 60 min to degrade dextrin and glucan in the sample, and then cooled the sample to room temperature for next step.

##### Step Two

After step one, sample was adjusted to pH 4.50 with aqueous hydrochloric acid solution (HCl, 4 mol/L) and then incubated at 40 °C, added 600 μL of amyloglucosidase, mixed by a magnetic stirring for one hour to hydrolysis of terminal α-1, 4 and α-1, 6 D-glucose residues from non-reducing ends of maltodextrins.

##### Step Three

The sample after step two was adjusted to pH 6.50 with NaOH (6 mol/L) and then incubated at 60 °C by shaking for 60 min with 300 μL of protease adding. This step was to hydrolyse the protein including the enzymes added into the bottle above. Then the treated sample was placed into a boiling water bath for 10 min to denature proteins, cool to the room temperature and then separate the insoluble substances. The supernatant of B/BF/DB was recovered by centrifugation at 5000 rpm for 15 min (LXJ- IIB, Shanghai Anting Scientific Instrument Factory, China).

##### Step Four

The supernatant of B/BF/DB recovered was concentrated using a rotary evaporator (RE-52AA, Shanghai Yarong Biochemistry Instrument Factory, Shanghai, China) at 50 °C.

### 3.3. Fractionation of Arabinoxylan by Gradient Ethanol Precipitation

Four fractions of arabinoxylans were obtained using gradient ethanol precipitation. Ethanol was slowly added into sample solution with constant stirring at room temperature to a final concentration of 50% (*w*/*v*). The resulting solution was incubated at 4 °C overnight to facilitate the aggregation, and then the precipitates were collected by centrifugation (5000 rpm for 15 min). Thereafter the precipitates were dispersed in distilled water and concentrated to remove residual ethanol, then dried in a freezing-dryer. The fractions of 50% ethanol precipitation were designated as B50/BF50/DB50. The concentration of ethanol in the supernatant was increased stepwise to 67% and a final concentration of 80%. The corresponding fractions were designated as B67/BF67/DB67, B75/BF75/DB75, and B80/B80/DB80, respectively. The fractionation procedure was summarized in Figure 3 and repeated to collect enough samples for analysis. The procedure details were shown as follows.

Detailed analysis of B50/BF50/DB50, B67/BF67/DB67, B75/BF75/DB75, and B80/BF80/DB80 were carried out according to the following methods.

### 3.4. Analytical Methods

#### 3.4.1. Physico-Chemical Analysis

Protein content was determined by an automatic elemental analyzer (Rapid N exceed, Elementar Analysensysteme GmbH, Hanau, Germany), the factor of 5.83 was used to convert measured nitrogen into protein according to Dumas combustion method [36]. About 20 mg samples were separately prepared in aluminum foil (50 × 50 mm, Alfa Aesar, Fisher Scientific, Waltham, MA, USA) and covered tightly for analysis. Ash and moisture were examined by AOAC methods [32].

#### 3.4.2. Scanning Electron Microscopy (SEM)

The SEM assay on the powder of lyophilized polysaccharide fractions were conducted according to Iravani et al. [37]. The samples coated with palladium gold were observed and photographed (SEM, Supra 55, Zeiss, Germany) with an accelerating voltage of 5 kV and a magnification of 20,000.

#### 3.4.3. Analysis of Monosaccharide Composition and Polysaccharides

Analysis of monosaccharides was performed according to Englyst and Cummings [38] and Li et al. [9] with some modifications. Samples were completely dissolved in deionized water immersed in a stirred glycerin bath at 105 °C. The solution samples were treated with hydrolysis, reduction, derivatization and extraction step by step for monosaccharides analysis by gas-chromatography equipped with a FID detector (Shimadzu Corporation, Kyoto, Japan).

##### Hydrolysis

Total solution samples (4 mL) was hydrolysed with 0.7 mL of 2 mol/L trifluoroacetic acid at 100 °C for 3 h. After cooling, the hydrolysates were diluted with deionized water to 5 mL and centrifuged for 15 min (4000× *g*).

##### Reduction

A 1 mL aliquot of acid solution was transferred into 25 mL centrifuge tube with 0.3 mL ammonia solution and 0.3 mL sodium borohydride (40 °C, 1 h). After that, 0.4 mL acetic acid was added to terminate the reaction.

##### Derivatization

A 1 mL aliquot of supernatant was transferred into 25 mL centrifuge tube, and 0.5 mL 1-methylimidazole and 4.5 mL acetic anhydride were added to react for 10 min exactly. A 10 mL aliquot of deionized water was added to stop the reaction and then the reaction product was cooled in an ice-water bath.

##### Extraction

A 3 mL aliquot of dichloromethane was added into the cooled reaction solution to extract for 5 min and the lower organic phase was drawn into a 10 mL flask. The operation was repeated twice and the organic phase was combined. The extract was washed twice using 20 mL deionized water and then made up to 10 mL by adding dichloromethane for GC injection.

Additionally, to analyse the content of sugar with a reducing end, a reduction step should be performed prior to hydrolysis and derivatization. Calibration was performed with standard solutions of L-arabinose, D-glucose, D-xylose, D-mannose and D-galactose.

(2)$$\mathrm{The}\;\mathrm{content}\;\mathrm{of}\;\mathrm{monosaccharides}\;(\mathrm{mg}/g)\;=\;\frac{\mathrm{mass}\;\mathrm{of}\;\mathrm{monosaccharide}\;\left(\mathrm{mg}\right)}{\mathrm{mass}\;\mathrm{of}\;B/\;\mathrm{BF}/\;\mathrm{DB}\;\mathrm{powder}\;\left(g\right)}$$

##### Chromatographic Condition

The formed alditol acetates (2 μL) were separated on a polar column (30 m × 0.32 mm i.d., 0.2 μm; DM-2330, DIKMA, Beijing, China) in a gas chromatograph (GC-2010, Shimadzu, Kyoto, Japan) equipped with an autosampler, a splitter injection port (split ratio 1:14) and a flame ionization detector (Shimadzu Corporation, Kyoto, Japan). The separation temperature, injection temperature and detection temperature were 240, 250 and 260 °C, respectively. N_2_ was used as the carrier gas.

The contents (dry weight) of arabinoxylan (AX), arabinose-galactose polymers (AG), mannose polymers (MP), glucose polymer (GP), A/X and avDP were calculated using Equations (3)–(8), respectively:(3)AX = (arabinose + xylose)×0.88
(4)AG = (0.7×132/150 + 0.9)×galactose 
(5)MP = mannose×0.9
(6)GP = glucose×0.9
(7)avDP = (% Ara − 0.7×% Gal +% Xyl) /% reducing end Xyl
(8)A/X = (% Ara − 0.7×% Gal) /% Xyl 
where Ara referred to arabinose, Xyl was xylose, Gal was galactose, and Man was mannose.

The number of 0.88 included in the formula was the conversion coefficient of arabinoxylan and the sum of arabinose and xylose, and 0.9 was the conversion coefficient of mannan and mannose. The formulae also included a correction for galactose, because wheat contains arabinogalactan with an Ara/Gal ration of 0.7 [39]. The factors 132 and 150 in Equation (4) reflect the molecular massed of anhydropentose sugars and pentose sugars, respectively.

#### 3.4.4. Analysis of Residual β-Glucan and Dextrin

β-Glucan and dextrin were detected by enzymatic method according to Megazyme manual. Samples were completely dissolved in deionized water immersed in a stirred glycerin bath at 105 °C. The samples were prepared following the instructions.

#### 3.4.5. Molecular Characterization

Molecular weight distribution of purified arabinoxylan fractions were determined by gel permeation chromatography with multi-angle static light scattering (GPC-MALLS) [40]. GPC-MALLS measurement of polysaccharides was performed on a DAWN HELEOS-II multi-angle laser photometer (Wyatt Technology Corporation, California, USA), Optilab rEX Refractive Index Detector (Wyatt Technology Corporation, California, USA) with a LaChrom Elite pump L-2130 (Hitachi, Ltd., Japan) equipped with TSK-GEL G3000 PWxl and G4000 PWxl column (7.8 mm × 300 mm) for aqueous solution. Dextran from *Leuconostoc spp.* (Mr–40,000, Sigma-Aldrich., Shanghai, China) was injected as the standard twice.

The eluent was sodium chloride aqueous solution (NaCl, 0.1 mol/L) with a temperature of 25 °C and at a flow rate of 0.5 mL/min. All of the samples were dissolved in the eluent with a concentration of 1.0 mg/mL and filtrated with polytetrafluoroethylene syringe membrane filter (0.45 µm pore size, Fisher Scientific, Waltham, USA). The injection volume was 200 µL, and the elution time was 30 min. Astra software was utilized for data acquisition and analysis.

### 3.5. Data Analysis

The DPS software (Version 7.5, Enfield, UK) was used to analyse data statistically. The results were shown as mean values ± standard deviations. The confidence level for statistical significance was set at 95% (*p* = 0.05) using a Tukey test. The Sigmaplot software (Version 12.5, Systat Software, San Jose, USA) and Excel 2007 were used to plot the figures.

## 4. Conclusions

The differences of arabinoxylan in wheat beer, beer foam, and defoamed beer by gradient ethanol precipitation were investigated. More arabinoxylan accumulated in the beer foam featured by the largest value of avDP, especially in 50% ethanol precipitation fraction. The highest A/X value of arabinoxylan was also presented in all beer foam fractions regardless of the concentration of ethanol. Under the same ethanol concentration, the Mw of arabinoxylan showed the lowest values in the beer foam, followed by that in wheat beer and defoamed beer. Therefore, it can conclude that the arabinoxylan accumulated in beer foam with the highest A/X and avDP and it was speculated that arabinoxylan may have an important effect on the performance of beer foam. Further investigations will continue, particularly on its other structure and effects on wheat beer foam.

## Figures and Tables

**Figure 1 molecules-24-01230-f001:**
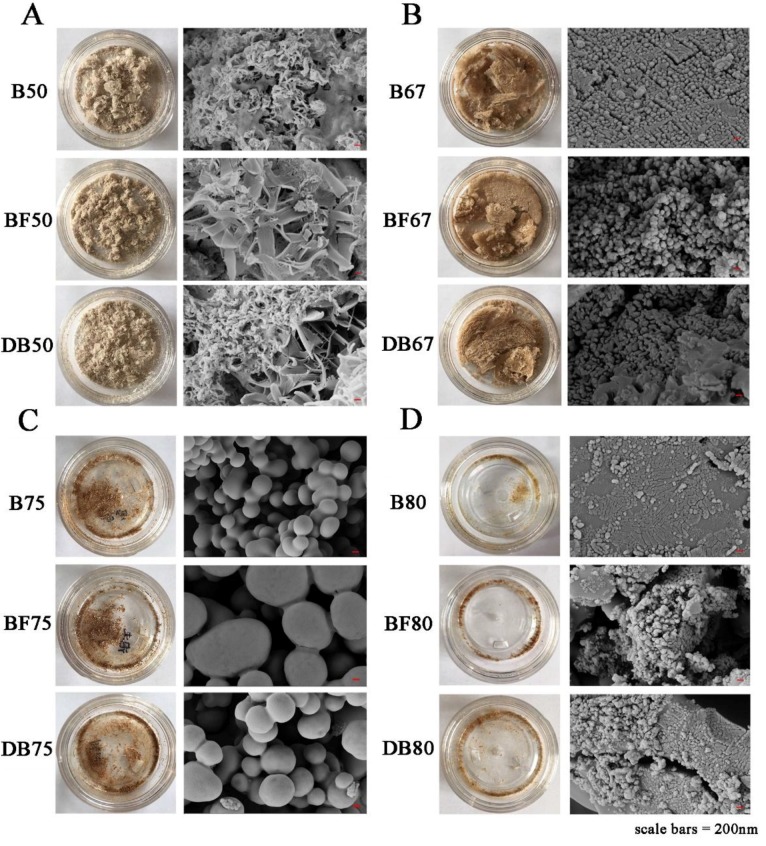
Photographs and SEM images of arabinoxylan. (**A**) referred to the photograph and SEM images of AX precipitated by 50% ethanol; (**B**) referred to the photograph and SEM images of AX precipitated by 67% ethanol; (**C**) referred to the photograph and SEM images of AX precipitated by 75% ethanol; (**D**) referred to the photograph and SEM images of AX precipitated by 80% ethanol.

**Figure 2 molecules-24-01230-f002:**
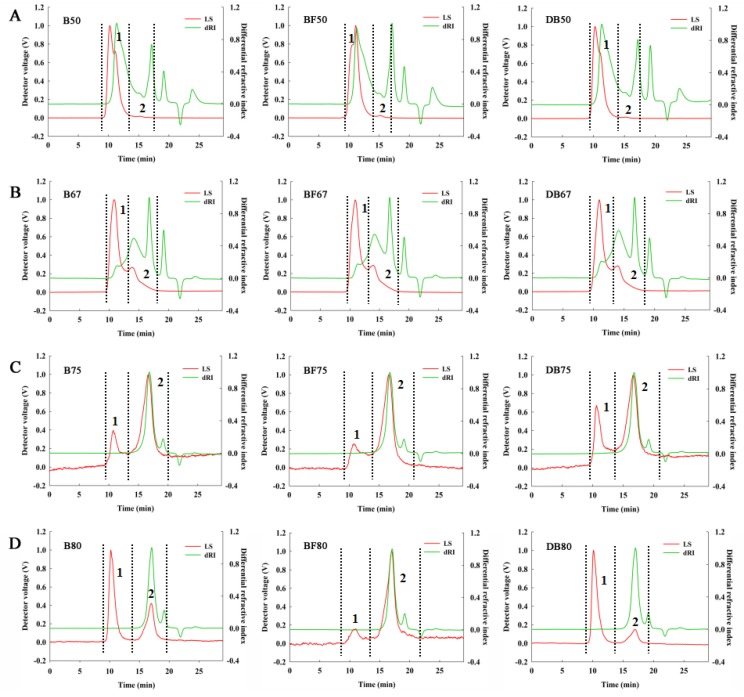
Distribution of molecular weight of arabinoxylan.

**Figure 3 molecules-24-01230-f003:**
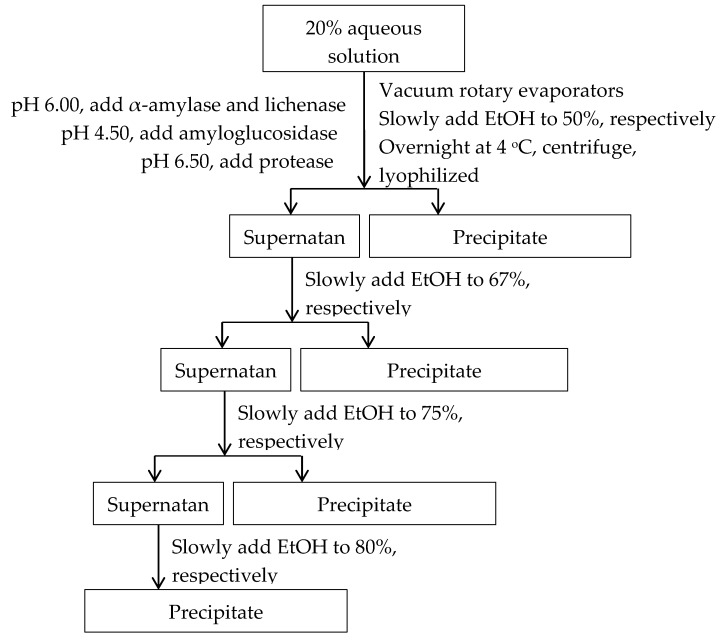
Procedure for arabinoxylan fraction by gradient ethanol precipitation.

**Table 1 molecules-24-01230-t001:** Yields and chemical compositions of freeze dried wheat beer and its fractions.

Fractions	B	BF	DB
Yield ^1^ (%)	31.88 ± 0.37 a	32.93 ± 0.72 a	31.83 ± 1.10 a
Protein (%)	14.2 ± 0.2 a	14.9 ± 1.1 a	14.9 ± 0.6 a
Moisture (%)	3.6 ± 0.0 c	4.4 ± 0.0 a	4.0 ± 0.0 b
Ash (%)	1.2 ± 0.1 a	1.0 ± 0.2 a	1.2 ± 0.0 a
Monosaccharides (mg/g)			
Arabinose	15.97 ± 0.25 b	16.99 ± 0.16 a	15.17 ± 0.51 b
Xylose	23.11 ± 0.22 a	22.66 ± 0.35 a	20.50 ± 0.91 b
Mannose	6.02 ± 0.49 a	6.23 ± 0.31 a	6.81 ± 1.27 a
Galactose	5.19 ± 0.67 a	5.16 ± 0.33 a	5.21 ± 0.80 a
Glucose	548.61 ± 3.10 b	587.84 ± 1.40 a	547.34 ± 3.35 b
Total sugar	598.90 ± 2.83 b	638.87 ± 1.17 a	595.03 ± 4.13 b

Notes: Beer (B), beer foam (BF), defoamed beer (DB). Yield ^1^ (%) = $\frac{\mathrm{weight}\;\mathrm{of}\;B/\;\mathrm{BF}/\;\mathrm{DB}\;\mathrm{powder}}{\mathrm{weight}\;\mathrm{of}\;B/\;\mathrm{BF}/\;\mathrm{DB}\;\mathrm{liquid}}\times 100\%$. Values within the same row by different letters are significantly different at *p* < 0.05 using a Tukey test.

**Table 2 molecules-24-01230-t002:** Analysis of yields, protein, and monosaccharides in ethanol precipitates.

Fractions	Yields^2^ (%)	Protein (%)	Monosaccharides^2^ (mg/g)				
			Arabinose	Xylose	Mannose	Galactose	Glucose
B50	12.56 ± 0.62 a	9.37 ± 0.66 a	117.39 ± 0.81 c	227.15 ± 1.10 c	97.33 ± 2.17 a	6.04 ± 0.47 b	-
BF50	13.22 ± 0.99 a	7.58 ± 1.22 a	151.94 ± 0.52 a	271.35 ± 1.48 a	87.37 ± 1.33 b	6.21 ± 0.71 b	-
DB50	14.01 ± 0.36 a	6.11 ± 070 a	141.55 ± 2.10 b	257.56 ± 3.18 b	89.28 ± 1.74 b	8.43 ± 1.53 a	-
B67	20.15 ± 0.97 a	10.31 ± 1.30 a	115.91 ± 1.50 a	111.31 ± 1.97 b	9.54 ± 0.62 a	55.86 ± 1.01 a	15.37 ± 1.56 b
BF67	20.08 ± 0.59 a	8.38 ± 0.17 a	109.16 ± 0.43 c	112.98 ± 0.66 ab	10.65 ± 0.38 a	42.61 ± 1.48 c	23.72 ± 1.65 a
DB67	20.85 ± 0.73 a	7.94 ± 0.05 a	113.27 ± 0.33 b	115.56 ± 0.57 a	9.22 ± 1.80 a	49.44 ± 1.55 b	17.48 ± 3.27 b
B75	19.65 ± 0.16 a	5.23 ± 0.28 b	33.96 ± 0.42 c	43.47 ± 1.99 c	11.83 ± 1.13 a	7.09 ± 0.07 c	197.35 ± 3.07 b
BF75	19.85 ± 0.83 a	5.48 ± 0.52 b	45.40 ± 1.03 a	57.27 ± 0.34 a	15.08 ± 1.82 a	8.90 ± 0.77 b	193.10 ± 2.66 b
DB75	18.32 ± 0.62 a	6.98 ± 0.11 a	41.43 ± 0.66 b	51.60 ± 0.73 b	13.52 ± 1.03 a	11.61 ± 0.83 a	206.96 ± 0.79 a
B80	7.23 ± 0.16 a	3.86 ± 0.17 a	23.54 ± 0.82 b	31.95 ± 1.51 a	20.47 ± 1.64 a	4.43 ± 1.57 a	167.89 ± 1.22 b
BF80	7.29 ± 0.65 a	7.91 ± 1.16 a	22.60 ± 0.48 b	31.00 ± 0.21 a	13.24 ± 0.33 b	4.41 ± 0.43 a	174.03 ± 1.37 a
DB80	7.88 ± 0.17 a	4.83 ± 1.30 a	26.16 ± 1.08 a	31.83 ± 2.59 a	16.10 ± 3.57 ab	6.04 ± 1.22 a	148.51 ± 1.88 c

Notes: Yields^2^ are expressed as weight percentage of B/BF/DB powder (as dry basis). Values within the same parts of ethanol precipitate by different letters are significantly different at *p* < 0.05 using a Tukey test. Beer (B), beer foam (BF), defoamed beer (DB), 50% ethanol precipitate (B50/BF50/DB50), 67% ethanol precipitate (B67/BF67/DB67), 75% ethanol precipitate (B75/BF75/DB75), 80% ethanol precipitate (B80/BF80/DB80). “-” indicates that it is not detected by GC-FID.

**Table 3 molecules-24-01230-t003:** Analysis of polysaccharides in wheat beer, beer foam, and defoamed beer.

Fractions	AXP			AG (mg/g)	MP (mg/g)	GP (mg/g)			Total GP (mg/g)
	AX (mg/g)	A/X	avDP			β-Glucan	Dextrin	Other GP	
B	34.39 ± 0.31 b	0.53 ± 0.03 a	24.90 ± 0.54 a	7.87 ± 1.01 a	5.42 ± 0.44 a	13.13 ± 0.25 b	0.53 ± 0.03 a	479.11 ± 2.79 b	493.75 ± 2.79 b
BF	34.89 ± 0.44 a	0.59 ± 0.01 a	20.48 ± 0.35 b	7.82 ± 0.50 a	5.61 ± 0.28 a	12.99 ± 0.14 b	0.59 ± 0.01 a	514.57 ± 1.26 a	529.05 ± 1.26 a
DB	31.39 ± 1.10 b	0.56 ± 0.04 a	17.01 ± 0.86 c	7.90 ± 1.22 a	6.13 ± 1.14 a	14.10 ± 0.28 a	0.56 ± 0.04 a	476.84 ± 3.02 b	492.60 ± 3.02 b
B50	303.19 ± 1.63 c	0.50 ± 0.00 c	961.58 ± 5.67 b	9.16 ± 0.71 a	87.60 ± 1.95 a	0.09 ± 0.01 a	0.50 ± 0.00 c	-	-
BF50	372.49 ± 1.54 a	0.54 ± 0.00 a	1065.22 ± 3.83 a	9.41 ± 1.07 a	78.64 ± 1.20 b	0.04 ± 0.00 b	0.54 ± 0.00 a	-	-
DB50	351.22 ± 4.64 b	0.53 ± 0.00 b	625.11 ± 1.62 c	12.78 ± 2.32 a	80.35 ± 1.56 b	0.05 ± 0.01 b	0.53 ± 0.00 b	-	-
B67	199.95 ± 3.03 ab	0.69 ± 0.01 a	49.79 ± 0.33 c	84.69 ± 1.52 a	8.59 ± 0.56 a	0.21 ± 0.02 b	0.69 ± 0.01 a	12.46 ± 1.40 b	13.83 ± 1.40 b
BF67	195.49 ± 0.96 b	0.70 ± 0.01 a	60.29 ± 1.01 a	64.60 ± 2.24 c	9.59 ± 0.35 a	2.08 ± 0.06 a	0.70 ± 0.01 a	18.56 ± 1.48 a	21.34 ± 1.48 a
DB67	201.37 ± 0.79 a	0.68 ± 0.01 a	54.46 ± 0.74 b	74.95 ± 2.35 b	8.30 ± 1.62 a	0.23 ± 0.01 b	0.68 ± 0.01 a	14.86 ± 2.95 ab	15.73 ± 2.95 b
B75	68.14 ± 2.11 c	0.67 ± 0.02 ab	58.60 ± 1.26 c	10.75 ± 0.11 c	10.65 ± 1.02 a	0.45 ± 0.04 b	0.67 ± 0.02 ab	175.43 ± 2.76 b	177.61 ± 2.76 b
BF75	90.35 ± 1.21 a	0.68 ± 0.00 a	89.96 ± 1.89 a	13.49 ± 1.17 b	13.57 ± 1.64 a	0.60 ± 0.05 b	0.68 ± 0.00 a	171.42 ± 2.39 b	173.79 ± 2.39 b
DB75	81.87 ± 1.19 b	0.65 ± 0.02 b	68.37 ± 1.87 b	17.60 ± 1.26 a	12.17 ± 0.93 a	3.69 ± 0.21 a	0.65 ± 0.02 b	180.76 ± 0.71 a	186.26 ± 0.71 a
B80	48.83 ± 1.69 a	0.64 ± 0.02 a	38.23 ± 1.25 b	6.71 ± 2.38 a	18.42 ± 1.47 a	9.04 ± 0.47 b	0.64 ± 0.02 a	139.50 ± 1.10 a	151.11 ± 1.10 b
BF80	47.17 ± 0.26 a	0.63 ± 0.02 a	49.13 ± 0.60 a	6.68 ± 0.64 a	11.91 ± 0.30 c	11.27 ± 0.53 a	0.63 ± 0.02 a	142.81 ± 1.24 a	156.63 ± 1.24 a
DB80	51.03 ± 3.19 a	0.69 ± 0.05 a	50.21 ± 1.81 a	9.16 ± 1.86 a	14.49 ± 0.21 b	7.45 ± 0.38 c	0.69 ± 0.05 a	123.72 ± 1.69 b	133.66 ± 1.69 c

Note: AXP refers to arabinose-xylose polymers, AG refers to arabinose-galactose polymers, MP refers to mannose polymers, GP refers to glucan and dextrin. β-Glucan and dextrin are detected by enzymic method of Megazyme. “-” indicates that it is not detected by GC-FID. Beer (B), beer foam (BF), defoamed beer (DB), 50% ethanol precipitate (B50/BF50/DB50), 67% ethanol precipitate (B67/BF67/DB67), 75% ethanol precipitate (B75/BF75/DB75), 80% ethanol precipitate (B80/BF80/DB80). Values within the same parts of ethanol precipitate by different letters are significantly different at *p* < 0.05 using a Tukey test.

**Table 4 molecules-24-01230-t004:** Total AX and ratio in BF vs. DB after gradient ethanol.

	B_50-80_	BF_50-80_	DB_50-80_
Total AX (mg/g)	620.11 ± 1.58 c	705.49 ± 2.27 a	685.48 ± 7.32 b
AX_BF_: AX_DB_	1.03 ± 0.01		
Total AG (mg/g)	111.30 ± 3.96 a	94.18 ± 3.46 b	114.49 ± 5.70 a
AG_BF_: AG_DB_	0.82 ± 0.05		
Total MP (mg/g)	125.26 ± 2.29 a	113.71 ± 2.40 b	115.31 ± 4.24 b
MP_BF_: MP_DB_	0.99 ± 0.02		
Total GP (mg/g)	340.31 ± 6.19 a	350.40 ± 3.79 a	334.31 ± 2.18 a
GP_BF_: GP_DB_	1.05 ± 0.01		

Note: B_50-80_/BF_50-80_/DB_50-80_ referred to the total polysaccharides in B/BF/DB under the 50–80% ethanol.

**Table 5 molecules-24-01230-t005:** Analysis of molecular weight of arabinoxylan.

Fractions (Da)	Peak 1				Peak 2				Peak1:Peak2
	Mn	Mp	Mw	Mw/Mn	Mn	Mp	Mw	Mw/Mn	
B50	124,200	301,000	482,400	3.9	32,900	30,700	34,800	1.0	94.56:5.44
BF50	74,350	245,200	472,200	3.1	49,970	47,440	51,470	1.0	93.56:6.44
DB50	88,120	300,800	496,000	5.6	33,880	37,320	44,350	1.0	92.64:7.36
B67	90,210	41,800	243,000	2.7	19,940	22,970	21,770	1.1	23.40:76.60
BF67	98,040	45,290	233,000	2.4	18,270	21,940	20,790	1.1	21.54:78.46
DB67	88,800	41,470	203,200	2.3	20,460	23,870	22,260	1.1	25.55:74.46
B75	364,500	237,800	558,800	1.5	3176	2452	4514	1.4	0.26:99.76
BF75	-	-	-	-	3293	2329	3275	1.6	1.40:98.60
DB75	46,580	26,480	91,920	2.0	2794	2288	5163	1.2	1.93:98.07
B80	816,200	673,000	198,300	2.4	1435	1238	1727	1.2	0.19:99.81
BF80	-	-	-	-	1704	1821	2798	1.2	0:100.00
DB80	573,500	524,700	225,200	3.9	1395	1166	2110	1.5	0.61:99.39

Note: “-” indicates that it is not analysed by Astra software. Mn: number-average molecular weight; Mp: peak molecular weight; Mw: weight-average molecular weight. 50% ethanol precipitate (B50/BF50/DB50), 67% ethanol precipitate (B67/BF67/DB67), 75% ethanol precipitate (B75/BF75/DB75), and 80% ethanol precipitate (B80/BF80/DB80).

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
