# Peer review of "Molecular Characterization of Arabinoxylan from Wheat Beer, Beer Foam and Defoamed Beer"

_molecules, 2019, doi:10.3390/molecules24071230_

Round 1
Reviewer 1 Report
Wang and coworkers explore the distribution of arabinoxylan in wheat beer and its foam. For this they use the fractionation of the beer, beer foam and defoamed beer by means of ethanol precipitation. They also offer molecular characterisation by GPC, SEM and substitution degree through sugar analysis by GC-FID. The work is scientifically sound and will be of interest to the readers of molecules. Some minor clarifications in the text would however help the reader -below some minor remarks that the authors might consider for further improving the manuscript:
- The yield of the B, BF and DB fraction appears to be quite low as it is the basis for further fractionation. A comment on the calculation basis of the yield and the composition of the rest of the fraction would help the reader.
- B, BF and DB should be somehow dependant variables as DB + BF = B. Therefore when BF is high DB should be lower or at least within the error of B.
- Are materials in SEC fully recovered or is full recovery just assumed? Specifying a recovery rate would support the discussion.
- In line 264, 265, 267 the units seem to be incorrectly selected
- Please add units in table 3
Author Response
Dear Editor and Reviewer:
Thank you very much for your careful review and constructive suggestions with regards to our manuscript “Molecular Characterization of Arabinoxylan from Wheat Beer, Beer Foam and Defoamed Beer” (molecules-461065). All comments are valuable and helpful for revising and improving our paper. We have studied the comments carefully and tried our best to revise and improve the manuscript. Revised portion is marked in red in the manuscript. In addition, we have also carried on the detailed revision to the grammar and language in order to meet the requirements of Molecules. The main corrections in manuscript and the responds to the reviewers’ comments are as follows. We earnestly appreciate your warm work, and hope that the corrections will meet with approval. Please feel free to contact us with any questions and we are looking forward to your reconsideration.
Sincerely yours,
Jinhua Du
Responds to the reviewer 1 comments:
Point 1: The yield of the B, BF and DB fraction appears to be quite low as it is the basis for further fractionation. A comment on the calculation basis of the yield and the composition of the rest of the fraction would help the reader.
Response 1: Thanks for your review and suggestion. The yield of the B, BF and DB fraction referred to the ratio of B/ BF/ DB powder to B/ BF/ DB liquid after freeze-drying. Arabinoxylan was extracted from dry powder rather than liquid beer in the manuscript. We had added supplementary explanation below Table 1 and in the part of 3.2.1.
Yield (%)=
Point 2: B, BF and DB should be somehow dependant variables as DB + BF = B. Therefore when BF is high DB should be lower or at least within the error of B.
Response 2:Thanks for your review. As you commented in theory, B/ BF/ DB should be somehow dependant variables as DB + BF = B. But in fact, B/ BF/ DB was regard as an independent material to suffer same treatment and subsequent analysis in our manuscript, besides, there were different extraction rate and loss among the three parts. Therefore, DB + BF ≠ B based on the results in the manuscript.
Point 3: Are materials in SEC fully recovered or is full recovery just assumed? Specifying a recovery rate would support the discussion.
Response 3: Thanks for your review. The recovery rate in SEC referred to the ratio of the real mass calculated by software to the theoretical mass, which can be used to determine the solubility of the sample. In our manuscript, the recovery rate of all materials in SEC was assumed to be fully recovered. We had added some discussion about the recovery rate in our manuscript.
Point 4: In line 264, 265, 267 the units seem to be incorrectly selected.
Response 4: Thank you for your careful review and we had already correct the mistakes in Line 264, 265, 267, as well as other similar incorrect units (Line 268, 269 and 276). The sentences had been changed as “In B50/ BF50/ DB50, it is shown as 94.56 %, 93.56 % and 92.64 % of total mass after analyzing with Mw of 482.40 kDa, 472.20 kDa and 496.00 kDa, respectively. In the precipitates of 67 % ethanol, two peaks signal can be separated for 243.00 kDa, 233.00 kDa and 203.20 kDa with the proportion of 23.40 %, 21.54 % and 25.55 %, and another peaks of 21.77 kDa, 20.79 kDa and 22.26 kDa, which are associated with the proportion of 76.60 %, 78.46 % and 74.46 %. In addition, B75/ BF75/ DB75 possesses more than 98% mass fraction of the total fraction and Mws are 4.514 kDa, 3.275 kDa and 5.163 kDa, respectively.”.
Point 5: Please add units in table 3.
Response 5: Thanks for your suggestion and we had added units (mg/g) in table 3.

Reviewer 2 Report
This work primarily shows arabinoxylan contents and composition in beer fractions after defoaming. Molecular weight analysis and SEM were carried out. The manuscript also presents content and composition data on other co-precipitated polysaccharides from ethanol extractions – arabinogalactans and mannans. While the manuscript contains useful data, the follow comments need to be addressed to allow for publication.
“about 1.5 ~ 2.5% of arabinoxylans could be found in the endosperm, where they constituted 66% of the endosperm cell wall.” These percentages are not clear. How is that when there are 66% in the cell wall, they are 1.5-2.5%?
“Wheat beer was made as previously described in Shandong Taishan Beer Co. Ltd. China” Please mention the % wheat in beer fermentation and any other details relevant to this work. Is it wheat-only beer or wheat-barley mix? What would be the estimated AX% in the wheat used for brewing?
How was defoaming of beer carried out?
“Enzymolysis was conducted according to the Megazyme manuals.” Please add weblink citations referring to the Megazyme manuals.
“was removed from the powder of B/ BF/ DB by n-hexane for three times according to” By Soxhlet?
“to pH = 6.00 by sodium hydroxide solution (NaOH, 6 mol/L).” What was the pH before neutralization?
“containing 600 μL of α-amylase and 300 μL of lichenase in a 60 oC water bath for 60” Were these in buffer? At what pH? Please describe the methods properly and check the manuscript again for such cases. They were in 100 mL bottles, but how much was the liquid volume in these bottles? What was the solid%? Were the bottles shaken?
Please cite the original papers for the methods, as appropriate.
Please define equation for yield in methods section. Are the yields of dry mass of powders obtained after rotavap and freeze drying?
Monosaccharides mg/g. What is the basis? Define in methods section.
Monosaccharide yields: Are these monomer yields or corrected for anhydrous correction?
“Glucose has the highest content in raw material of B/ BF/DB” Is the glucose monomeric glucose or mainly in glucan form? I believe it is glucan as ethanol precipitations would remove any monomeric glucose. But in the introduction section, you had written that the arabinoxylans are the polysaccharides. However, Table 1 shows glucan is very high. Where does the glucan come from?
Table 2: “Yields are expressed as weight percentage (as dry basis).” Weight percentage of what? Total powder? Hemi in powder?
“Therefore, we could know that arabinoxylan is mainly collected in the 50 % and 67 % ethanol precipitates and it is more easily accumulated in beer foam according to the highest value of A/X and avDP.” Table 3 doesn’t show big difference in AX between foam and defoamed fractions. If you go strictly by the posthoc test, DB67 actually shows higher AX% than BF67. Then how are these statements that AX accumulates more in form valid? Moreover, I don’t think that there is a significant difference between 195 and 201. Also, how can you assess accumulation from in different fractions from average degree of polymerization?
Please write units for Table 3 in the header row. Are these percentages?
What was the dn/dc for MALLS and how was it determined?
It is not clear which one is Peak 2 in Fig 2A for 50% ethanol. Is it the shoulder?
“beer and defoamed beer. Therefore, it can conclude that the arabinoxylan accumulated in beer foam would present important influence in the” I don’t agree with the conclusion statement based on data in Table 3.
What is the significance of higher AX molecular weight in beer foam?
I did not clearly understand the difference between B, BF and DB. By B. Did you mean that the results for B are for whole beer (total) without any defoaming? If yes, then sugars in B are total sugars, and you could have calculated polysaccharide ratio in BF vs. DB. Like AX ratio in BF vs. DB and total AX in B. Why was the B, BF and DB format chosen, as it would have been more interesting to learn about the percentages in BF and DB as part of total polysaccharide in B?
Why do you have the order as Introduction, Results and Discussion, Methods, and Conclusions. This looks unusual as usually Methods are before Results.
Author Response
Dear Editor and Reviewer:
Thank you very much for your careful review and constructive suggestions with regards to our manuscript “Molecular Characterization of Arabinoxylan from Wheat Beer, Beer Foam and Defoamed Beer” (molecules-461065). All comments are valuable and helpful for revising and improving our paper. We have studied the comments carefully and tried our best to revise and improve the manuscript. Revised portion is marked in red in the manuscript. The main corrections in manuscript and the responds to the reviewers’ comments are as follows. We earnestly appreciate your warm work, and hope that the corrections will meet with approval. Please feel free to contact us with any questions and we are looking forward to your reconsideration.
Sincerely yours,
Jinhua Du
Responds to the reviewer’ comments:
Reviewer 2:
Point 1: “about 1.5 ~ 2.5% of arabinoxylans could be found in the endosperm, where they constituted 66% of the endosperm cell wall.” These percentages are not clear. How is that when there are 66% in the cell wall, they are 1.5-2.5%?
Response 1: Thanks for your review and we are so sorry for the vague expression. We looked up the references again and changed the sentence as “about 1.5 ~ 2.5% of arabinoxylans could be found in the endosperm, while arabinoxylan is one of the main non-starchy polysaccharides of the cell walls, which account for about 66% of total polysaccharide in endosperm cell walls.”
Point 2: “Wheat beer was made as previously described in Shandong Taishan Beer Co. Ltd. China” Please mention the % wheat in beer fermentation and any other details relevant to this work. Is it wheat-only beer or wheat-barley mix? What would be the estimated AX% in the wheat used for brewing?
Response 2: Thanks for your review and kindly suggestion. The detailed information was added in the part of Material and it is as follows:
“Wheat beer was made as previously described in our research (Li et al., 2017). Barley malt (0.51% WEAX ) and wheat malt (1.08% WEAX ) with the ratio of 1:1 were milled and mixed with water, the mixed mash then underwent a protein rest, saccharification, lautering, wort boiling, hopping and a cooling phase in sequence. Hop pellet was added at the beginning of boil. The boiled wort was separated from the hop waste and cooled to 20 oC and oxygenated. A 10 oP wort was transferred into a fermentation vessel, pitched with Saccharomyces cerevisiae WL 300 and fermented at 20 - 22 oC. Once the diacetyl level reached the limiting value, the green beer was immediately cooled to 0 oC and then entered maturation to wheat beer for analyze.”
Point 3: How was defoaming of beer carried out?
Response 3: Thank you for your review. The separation of defoaming beer referred to the previous method from Wu et al. (2017) with some modification, and now it has been supplemented in our manuscript. The details are as follows:
“A bottle of beer was poured into a separating funnel at a constant speed along the edge of funnel (diameter = 20 cm) to generate foam. The upper foam and lower liquid were collected as beer foam and defoamed beer, respectively. The sampled beer, beer foam and defoamed beer were centrifuged at 4500rpm for 15 min before further analysis.”
Point 4: “Enzymolysis was conducted according to the Megazyme manuals.” Please add weblink citations referring to the Megazyme manuals.
Response 4: Thanks for your review and suggestion. The weblink of the Megazyme manual had been added into the appropriate place in the manuscript.
Point 5: “was removed from the powder of B/ BF/ DB by n-hexane for three times according to” By Soxhlet?
Response 5: Thanks for your review and we are sorry for the incomplete description. It was not removed by Soxhlet. The details had been modified in the manuscript. “Fat was removed from the powder of B/ BF/ DB by vibrating extraction for three times with n-hexane (1:5, w/v) under the condition of 50 oC, 2h, 200rpm, and then the defatted sample of B/ BF/ DB was dried for enzymolysis.”
Point 6: “to pH = 6.00 by sodium hydroxide solution (NaOH, 6 mol/L).” What was the pH before neutralization?
Response 6: Thanks for your review. The pH of B/ BF/ DB was 4.53/ 4.56/ 4.56 before neutralization and we had added this data in our manuscript.
Point 7: “containing 600 μL of α-amylase and 300 μL of lichenase in a 60 oC water bath for 60” Were these in buffer? At what pH? Please describe the methods properly and check the manuscript again for such cases. They were in 100 mL bottles, but how much was the liquid volume in these bottles? What was the solid%? Were the bottles shaken?
Response 7: Thank you for your review. We had checked our manuscript carefully and modified the relevant methods. The modified methods were as follows.
Step One: A 20 % aqueous solution of B/ BF/ DB was made with the original pH of 4.53/ 4.56/ 4.56, and then sodium hydroxide solution (NaOH, 6 mol/L) was used to adjust pH to 6.00 for enzymolysis. A 100 mL aqueous solution of B/ BF/ DB in a 500 mL bottle was incubated in a 60 oC water bath, then 600 μL of α-amylase and 300 μL of lichenase were added into the bottle by shaking for 60 min to degrade dextrin and glucan in the sample, and then cooled the sample to room temperature for next step.
Step Two: After step one, sample was adjusted to pH 4.50 with aqueous hydrochloric acid solution (HCl, 4 mol/L) and then incubated at 40 oC, added 600 μL of amyloglucosidase, mixed by a magnetic stirring for one hour to hydrolysis of terminal α-1, 4 and α-1, 6 D-glucose residues from non-reducing ends of maltodextrins.
Step Three: the sample after step two was adjusted to pH 6.50 with NaOH (6 mol/L) and then incubated at 60 oC by shaking for 60 min with 300 μL of protease adding. This step was to hydrolyze the protein including the enzymes added into the bottle above. Then the treated sample was placed into a boiling water bath for 10 min to denature proteins, cool to the room temperature and then separate the insoluble substances. The supernatant of B/ BF/ DB was recovered by centrifugation at 5000 rpm for 15 min (LXJ- IIB, Shanghai Anting Scientific Instrument Factory, China).
Step Four: The supernatant of B/ BF/ DB recovered was concentrated using a rotary evaporator at 50 oC.
Point 8: Please cite the original papers for the methods, as appropriate.
Response 8: Thanks for your review. We had checked and ensured that the original papers for method had been cited.
Point 9: Please define equation for yield in methods section. Are the yields of dry mass of powders obtained after rotavap and freeze drying?
Response 9: Thanks for your review and suggestion. The yield of the B, BF and DB fraction referred to the ratio of B/ BF/ DB powder after rotavap and freeze-drying to original B/ BF/ DB liquid. We had added supplementary explanation below Table 1 and in the methods section of 3.2.1.
Yield (%) =
Point 10: Monosaccharides mg/g. What is the basis? Define in methods section.
Response 10: Thanks for your review and suggestion. We have added the calculation of monosaccharides (mg/g) in methods section. The detail was as follows.
The content of monosaccharides (mg/g) =
Point 11: Monosaccharide yields: Are these monomer yields or corrected for anhydrous correction?
Response11: Thanks for your review. The monomer yields showed in this manuscript had already been corrected for dehydration.
Point 12: “Glucose has the highest content in raw material of B/ BF/ DB” Is the glucose monomeric glucose or mainly in glucan form? I believe it is glucan as ethanol precipitations would remove any monomeric glucose. But in the introduction section, you had written that the arabinoxylans are the polysaccharides. However, Table 1 shows glucan is very high. Where does the glucan come from?
Response 12: Thanks for your review. Table 1 showed the monosaccharides content in raw material of B/ BF/DB which was not treated by the ethanol. The high content of glucose actually referred to the sum of monomer glucose and glucose hydrolyzed from glucan and dextrin because they were determined by GC-FID. We have made some explanation in the appropriate places in the article.
Point 13: Table 2: “Yields are expressed as weight percentage (as dry basis).” Weight percentage of what? Total powder? Hemi in powder?
Response 13: Thanks for your review. We have modified the expression in the manuscript. Yields are expressed as weight percentage of B/BF/DB powder (as dry basis).
Point 14: “Therefore, we could know that arabinoxylan is mainly collected in the 50 % and 67 % ethanol precipitates and it is more easily accumulated in beer foam according to the highest value of A/X and avDP.” Table 3 doesn’t show big difference in AX between foam and defoamed fractions. If you go strictly by the posthoc test, DB67 actually shows higher AX% than BF67. Then how are these statements that AX accumulates more in form valid? Moreover, I don’t think that there is a significant difference between 195 and 201. Also, how can you assess accumulation from in different fractions from average degree of polymerization?
Response 14: Thanks for your review. We had checked and reconsidered the sentences, there was something wrong with this comment. Average degrees of polymerization referred to the polymerization of arabinoxylan in this manuscript and it is an important factor to evaluate the structural characteristics of AX, as well as A/X. Actually, average degree of polymerization cannot be used to access accumulation, however, the AX with higher avDP migrated more easily to the foam section. Therefore we changed this comment on the basis of your question and suggestions. The revised sentence is as follows.
“Therefore, we could know that arabinoxylan is mainly collected in the 50 % ethanol precipitates with the highest value of A/X and avDP, especially in beer foam.”
Point 15: Please write units for Table 3 in the header row. Are these percentages?
Response 15: Thanks for your suggestion and we had added units (mg/g) in table 3.
Point 16: What was the dn/dc for MALLS and how was it determined?
Response 16: Thanks for your review. dn/dc referred to the ratio of signal intensity to the concentration of dilute solution. The samples was subjected to a series of dilutions which the concentration should be known and accurate, after that the diluted solution was injected into the detector and the data was calculated by DNDC software automatically. But actually, dn/dc was usually needed when the absolute molecular weight was analyzed. In our manuscript, dextran from Leuconostoc spp. (Mr ~ 40,000, Sigma-Aldrich., Shanghai, China) was injected as the standard, and the data in table 4 referred to the relative molecular weight, so we didn’t presented the value of dn/dc in our manuscript which was 0.135 showed in test report.
Point 17: It is not clear which one is Peak 2 in Fig 2A for 50% ethanol. Is it the shoulder?
Response 17: Thanks for your review. Peak 2 in Fig 2A for 50% ethanol was a peak which was a little bit small beside the Peak 1, we had already marked peak 1 and peak 2 clearly and detailed in Fig.2.
Fig.2. Distribution of molecular weight of arabinoxylan
Point 18: “beer and defoamed beer. Therefore, it can conclude that the arabinoxylan accumulated in beer foam would present important influence in the” I don’t agree with the conclusion statement based on data in Table 3.
Response 18: Thanks for your review. We have reconsidered and reanalyzed the data in Table 3, and made a change of the conclusion. The detail was as follows and also presented in the appropriate part of the manuscript.
“... beer and defoamed beer. Therefore, it can conclude that the arabinoxylan accumulated in beer foam with the highest A/X and avDP and it was speculated that arabinoxylan may have an important effect on the performance of beer foam.”
Point 19: What is the significance of higher AX molecular weight in beer foam?
Response 19: Thanks for your review and this question also give us more inspiration for the research. The weight-average molecular weight (Mw) of AX in beer foam presented the lowest values, followed by that in wheat beer and defoamed beer. However, the avDP also could be used to evaluate the molecular weight of AX, the molecular weight of AX was higher in beer foam from the perspective of avDP. Therefore, we just speculate that AX which was extract from beer foam would have an influence on the beer foam performance.
Point 20: I did not clearly understand the difference between B, BF and DB. By B. Did you mean that the results for B are for whole beer (total) without any defoaming? If yes, then sugars in B are total sugars, and you could have calculated polysaccharide ratio in BF vs. DB. Like AX ratio in BF vs. DB and total AX in B. Why was the B, BF and DB format chosen, as it would have been more interesting to learn about the percentages in BF and DB as part of total polysaccharide in B?
Response 20: Thanks for your review and these constructive suggestion. The results showed of B were for whole beer without any defoaming in our manuscript. B, BF and DB format were chosen because we thought that arabinoxylan, as an important non-starch polysaccharide, would have effects on beer foam performance. Thus, we divided the beer into beer foam and defoamed beer to explore the structural characteristics and distribution of AX in these different parts. We had calculated the AX ratio in BF vs. DB and total AX in B. The relevant data were shown below and we had made some explanation in the manuscript.
Table 4 Total AX and ratio in BF vs. DB after gradient ethanol
B | BF | DB | |
Total AX (mg/g) | 620.11±1.58 | 705.49±2.27 | 685.48±7.32 |
AXBF: AXDB | 1.03±0.01 | ||
Total AG (mg/g) | 111.30±3.96 | 94.18±3.46 | 114.49±5.70 |
AGBF: AGDB | 0.82±0.05 | ||
Total MP (mg/g) | 125.26±2.29 | 113.71±2.40 | 115.31±4.24 |
MPBF: MPDB | 0.99±0.02 | ||
Total GP (mg/g) | 340.31±6.19 | 350.40±3.79 | 334.31±2.18 |
GPBF: GPDB | 1.05±0.01 |
Note: B50-80/ BF50-80/ DB50-80 referred to the total polysaccharides in B/ BF/ DB under the 50% - 80% ethanol.
Point 21: Why do you have the order as Introduction, Results and Discussion, Methods, and Conclusions. This looks unusual as usually Methods are before Results.
Response 21: Thanks for your review. According to the Molecules Microsoft Word template file, the order for the article is Introduction, Results and Discussion, Method and Conclusion. So, we made this order in our manuscript as you mentioned above.

Round 2
Reviewer 2 Report
Authors answered all the comments and carefully revised the manuscript. The revised manuscript is clearer and contains interesting findings on arabinoxylan yield and degree of polymerization in beer foam and defoamed beer. It can now be accepted for publication. However, the author list has changed which needs explanation depending on journal’s guidelines.
I’m not sure of the policy for ‘Molecules’ but may need to contact the editor and write a letter explaining why this was the case and consent from all authors that they agree with the changes.
Revised author list: Jie Li a, Jinhua Du a, *
Original Author List: Jie Li , Jinhua Du * , Qi Wang *
Author Response
Responses to review 2 Point 1: I’m not sure of the policy for ‘Molecules’ but may need to contact the editor and write a letter explaining why this was the case and consent from all authors that they agree with the changes. Revised author list: Jie Li a, Jinhua Du a, * Original Author List: Jie Li , Jinhua Du * , Qi Wang * Response 1: Thanks for your review. We are so sorry to change the authors’ information as Dr. Wang thought her contribution was not enough to act as a co-author. We have thanked Dr. Wang in the acknowledgement of the first revised manuscript for her help of experimental design, data analysis and English language editing. All the original authors will send an e-mail to the editor for confirming this change and attached is the confirmation copy signed by the new authors.